# Iodine Intake from Universal Salt Iodization Programs and Hashimoto’s Thyroiditis: A Systematic Review

**DOI:** 10.3390/diseases13060166

**Published:** 2025-05-23

**Authors:** Hernando Vargas-Uricoechea, Alejandro Castellanos-Pinedo, Ivonne A. Meza-Cabrera, María V. Pinzón-Fernández, Karen Urrego-Noguera, Hernando Vargas-Sierra

**Affiliations:** 1Metabolic Diseases Study Group, Department of Internal Medicine, Universidad del Cauca, Carrera 6 No 13N-50, Popayán 190001, Colombia; karenurrego@unicauca.edu.co (K.U.-N.); hdvargas@unicauca.edu.co (H.V.-S.); 2Faculty of Medicine, Universidad del Sinú, Hospital San Jerónimo, Montería 230001, Colombia; acaspinedo@yahoo.es; 3Metabolic Diseases Study Group, Department of Pathology, Universidad del Cauca, Carrera 6 No 13N-50, Popayán 190001, Colombia; imeza@unicauca.edu.co; 4Health Research Group, Department of Internal Medicine, Universidad del Cauca, Carrera 6 No 13N-50, Popayán 190001, Colombia; mpinzon@unicauca.edu.co

**Keywords:** iodine, salt, Hashimoto, prevalence, autoimmunity, thyroid

## Abstract

Background: Hashimoto’s thyroiditis (HT) is characterized by the loss of tolerance to thyroid autoantigens [thyroid peroxidase (TPO) and thyroglobulin (Tg)], usually identifying circulating antibodies (Abs) against these thyroid autoantigens (TPOAb and/or TgAb), together with a significant lymphocytic infiltration, causing an increased risk of hypothyroidism. Among the multiple mechanisms described for the development of HT is the nutritional status of several micronutrients, including iodine. Iodine deficiency or excess is associated with thyroid function disorders and, likely, thyroid autoimmunity. Thus, iodized salt intake [especially through universal salt iodization (USI) programs] may be influencing the prevalence of HT. The objectives of this systematic review are to describe and analyze changes over time in the prevalence of HT following the implementation of USI programs. Methods and results: The following databases were consulted for articles published from January 1965 to January 2025: Pubmed/Medline; ProQuest; Scopus; Biosis; Web of Science; and Google Scholar. The search terms were as follows: “iodine”, “salt”, “intake”, “prevalence”, AND Hashimoto’s thyroiditis. Only English language articles were taken into account, and each of them was scrutinized according to the JBI Critical Appraisal Checklist. Only those studies in which the design, study population, number of participants, country, evaluation post-USI (years)*,* and the prevalence of thyroid Abs positivity were described were included. In total, 74 studies were identified, of which 31 evaluated thyroid Abs values post-USI. Conclusions: Excess iodine intake, mediated by USI programs without an adequate follow-up and monitoring plan, may explain (at least in part) the prevalence and distribution of HT; therefore, it is a real challenge to establish a balance between healthy salt intake, USI program strategies, and possible functional outcomes and thyroid autoimmunity in the population. Registration number: INPLASY202540074.

## 1. Introduction

Autoimmune thyroid diseases (AITDs) are a group of diseases characterized by the breakdown of tolerance to thyroid autoantigens [primarily thyroid peroxidase (TPO), thyroglobulin (Tg), and the thyrotropin receptor (TR)]. This induces both humoral and cellular autoimmune responses, ranging from the presence of circulating antibodies (Abs) to thyroid autoantigens (TPOAb, TgAb, and TRAb) and lymphocytic infiltration [1,2].

The two extremes of the clinical presentation of AITDs are hyperthyroidism [Graves–Basedow disease (GBD)] and hypothyroidism [Hashimoto’s thyroiditis (HT)]. HT is the leading cause of hypothyroidism in iodine-sufficient areas, with its widely variable prevalence being dependent on multiple factors. The estimated global prevalence is between 5 and 10%, with areas showing prevalences as high as >20% and as low as <0.5%. It is significantly higher in women than in men [3,4].

Numerous nutritional, genetic, environmental, and socioeconomic factors, inter alia, can affect the global prevalence of HT. Among nutritional factors, multiple studies have suggested that micronutrient deficiencies (especially those relating to iodine, selenium, zinc, iron, vitamin D, and vitamin B12, among others) are associated with increased thyroid Ab positivity.

Likewise, iodine is a micronutrient that is not only involved in thyroid function, but also in the pathogenesis of AITDs [5,6]. Iodine, as a component of thyroid hormones, represents 65% and 58% of the weight of thyroxine (T4) and triiodothyronine (T3), respectively; therefore, iodine status is directly associated with thyroid function, expressing a “U”-shaped relationship between iodine intake and the risk of thyroid diseases, e.g., an insufficient intake is associated with goiter, hypothyroidism, cretinism, adverse obstetric outcomes, and growth and intellectual development disorders, inter alia (iodine deficiency disorders—IDDs) [6,7,8].

In the presence of excessive iodine intake or exposure, hypothyroidism or hyperthyroidism may occur (when the intrinsic regulatory mechanisms of protection against excess iodine fail). Furthermore, sustained exposure to excess iodine has also been linked to multiple other outcomes; however, studies have not been able to reproduce these outcomes, including diabetes, hypertension, cardiovascular mortality, and papillary thyroid carcinoma, among others [6,7,8,9,10].

IDDs arise when iodine intake falls below the recommended levels. This term refers to each of the consequences of iodine deficiency in a given population that can be prevented, as long as an adequate intake of iodine is ensured, and remain a persistent public health problem worldwide, since they have a high clinical, social, and economic impact. It has been established that the most cost-effective strategy for their eradication and elimination is through the implementation of universal salt iodization (USI) programs [8,11,12].

This strategy has considerably reduced the health effects of IDDs in all countries where it has been implemented; however, the lack of monitoring and implementation of such programs has led to excess iodine consumption from salt (and other iodine-fortified foods) or a resurgence of IDDs in some geographic areas [11,12,13,14].

However, some studies have found that the prevalence of HT has increased in recent decades. It has been suggested that the lack of monitoring and control of USI (together with the increase in the consumption of iodized salt) are partially responsible for this discovery. On the contrary, other studies have shown that its prevalence has decreased slightly [10,13,14].

The objectives of this systematic review are to describe and analyze the change in the prevalence of HT, through the positivity of thyroid Abs (TPOAb and/or TgAb), in those areas where USI programs have been implemented.

## 2. Materials and Methods

This systematic review is part of a research project that determines and analyzes the distribution, associated factors, and behavior of AITDs in Colombia. This project was developed in accordance with the principles of the Declaration of Helsinki and has been approved by the Research Ethics Committee of the University of Cauca, Colombia (ID: 4656, January 2018). However, due to the nature of this study, Institutional Review Board approval was not required.

### 2.1. Literature Search and Selection Criteria

Using modified versions of the Population, Interventions, Comparators, and Outcomes (PICO) framework, we formulated the research question and selected the eligibility criteria for the systematic review (Table 1).

Subsequently, a structured literature search was carried out in PubMed/Medline, ProQuest, Scopus, Biosis, Web of Science, and Google Scholar for articles published from January 1965 to January 2025 (human trials, clinical trials, meta-analyses, reviews, scoping reviews, and systematic reviews). The search criteria were as follows: Iodine [Title/Abstract] OR Salt [Title/Abstract] OR Intake [Title/Abstract] OR Prevalence [Title/Abstract] AND Hashimoto’s Disease [Title/Abstract] OR Hashimoto’s Thyroiditis [Title/Abstract] OR Chronic Lymphocytic Thyroiditis [Title/Abstract] OR Autoimmune thyroiditis [Title/Abstract].

Only those studies in which the design, study population, number of participants, country and evaluation post-USI (years)*,* and the prevalence of thyroid Ab (TPOAb and/or TgAb) positivity were described were included. In total, 74 studies were included, of which 31 evaluated thyroid Ab values after USI programs (Figure 1).

### 2.2. Data Extraction

The titles and abstracts of all studies were independently reviewed by three investigators (H.V.-U., I.A.M.-C., and M.V.P.-F.) using the Rayyan web tool (this further helped reduce selection bias). Full texts for the studies that met the initial inclusion criteria were obtained and reviewed, and the data were extracted using a standardized template using a predefined data form created in Excel. In cases where discrepancies arose in the extracted data, the investigators collaboratively conducted a second round of extraction to validate the accuracy of the information. Each article was scrutinized according to the JBI Critical Appraisal Checklist. Letters, commentaries, preprints, letters to the editor, and non-peer-reviewed articles, as well as conference abstracts, were also excluded. Only articles written in English were considered (Table 2).

### 2.3. Data Analysis

The following data were collected: design, study population, number of participants, country, post-USI evaluation (years), and prevalence of TPOAb and/or TgAb positivity (95% CI). No statistical analysis or meta-analysis was performed due to the high heterogeneity observed among the studies included in this review. However, we developed a descriptive analysis to summarize and synthesize the most important characteristics of the selected studies (choosing to adopt a narrative approach).

This systematic review was registered on the “International Platform of Registered Systematic Review and Meta-Analysis Protocols INPLASY” (Registration number: INPLASY202540074; DOI: 10.37766/inplasy2025.4.0074) and followed the “Reporting Checklist for Systematic Reviews Based on the PRISMA guidelines” (Appendix A).

## 3. Results

### 3.1. Global Prevalence of HT

A total of 25 countries were found where the prevalence of HT had been assessed [Australia, Bosnia and Herzegovina, Brazil, China, Colombia, Croatia, Denmark, Finland, England (United Kingdom), Germany, Ghana, Iran, Italy, Japan, Jordan, South Korea, Mexico, Nigeria, Norway, Poland, Russia, Spain, Sri Lanka, Tunisia, and the USA] [3,4,15] (Table 3).

In Europe, prevalence rates as low as 0.42% and 0.43% have been reported in Russia and Bosnia and Herzegovina, respectively, and very high prevalence rates have been reported in Italy (35.5%) and Denmark (39.7%). Prevalence in Africa also varies, with the highest prevalence reported in Tunisia (22.8%) and the lowest in Nigeria (6.7%). In Oceania (Australia), prevalence ranges from 8.5% to 13%. In Asia, prevalence rates as low as 0.1% in South Korea and 0.3% in China have been reported. However, prevalence rates as high as 16.1% have also been reported in China, as well as in Japan (18%) and Jordan (15.1%). In the Americas, prevalences as low as 0.4% and 0.1% are found in the USA and Brazil, respectively, and prevalences as high as 22.4% and 22.3% have been reported in the USA and Colombia, respectively, which denotes the wide variability in the prevalence of HT depending on the geographic area studied.

### 3.2. Global Iodine Population Status

In a 2003 analysis, the WHO estimated that the Americas and Western Pacific regions had the lowest proportions of the population with insufficient iodine intake (9.8% and 24%, respectively). In the other regions, the figures were 56.9% (Europe), 54.1% (Eastern Mediterranean), 42.6% (Africa), and 39.8% (Southeast Asia). Additionally, it was found that between 1990 and 2003, the proportion of households consuming iodized salt increased from 10% to 66%. Consequently, the number of countries with IDDs decreased from 110 to 54 [18,19].

In 2006, 15 countries had reached the target of eliminating IDDs, and in 2008, the prevalence of iodine deficiency was calculated (excluding Western European countries and the USA); the highest prevalence was found in Europe (52%), followed by the Eastern Mediterranean (47.2%) and Africa (41.5%), with the lowest prevalence in descending order being in Southeast Asia, the Western Pacific, and the Americas (30%, 21.2%, and 11%, respectively) [19,20].

Between 2004 and 2023, the number of countries with excessive iodine intake increased (from 5 countries in 2004 to 15 countries in 2023). In 2023, 26 countries were still identified as reporting insufficient iodine intake (in the general population based on school-age children) [21,22].

Very recently, it was found that over the past 30 years, the prevalence rate of iodine deficiency among adolescents has decreased significantly, from 3082.43 (95% CI: 2473.01 to 3855.86) per 100,000 of the population to 2190.84 (95% CI: 1729.18 to 2776.16) per 100,000 of the population. Globally, the iodine deficiency prevalence among adolescents and young adults shows an oscillating pattern, decreasing between 1990 and 2000, with a subsequent increase between 2000 and 2009, and a continued decrease between 2010 and 2019 [23].

### 3.3. Population Iodine Status in Countries Where the Prevalence of HT Has Been Assessed

In each of the 25 countries in which the prevalence of HT has been described (Table 2), the iodine status of the population has also been assessed. Studies have also been conducted to investigate countries via an assessment of their median urinary iodine level (mUIC). It was found that four countries (Finland, Germany, Norway, and Russia) had insufficient iodine intake (mUIC < 100 μg/L); two had an excessive intake (Colombia and South Korea, with mUIC values ≥ 300 μg/L); and nineteen countries reported an adequate intake (mUIC values between 100 and 299 μg/L). The vast majority of these studies were conducted in school-age children (Table 4).

### 3.4. USI Programs, Population Iodine Status, and Thyroid Autoimmunity

A total of 31 studies (from 14 countries) have investigated the possible association between the implementation (mandatory) of USI programs and the prevalence of TPOAb and TgAb positivity. These studies evaluated the prevalence of TPOAb and TgAb positivity (at least during the first year from the time of implementation of the USI programs) (Table 5).

Following the implementation of the USI programs, the prevalence of TPOAb and/or TgAb positivity increased (or remained high) in eight countries (Sri Lanka, Greece, Denmark, Poland, Turkey, India, Italy, and Colombia); in four countries (Iran, Morocco, Brazil, and Germany), the prevalence decreased (or remained low); and in one country (Tasmania), the prevalence did not change (although the estimated baseline prevalence was high).

In China, the results were variable, with some areas where positivity increased and others where positivity decreased or remained unchanged over time [4,15].

## 4. Discussion

Although USI programs have significantly modified the prevalence of IDDs, the wide variability in relation to the prevalence of HT and other thyroid dysfunctions remains, both in areas where IDDs persist and in those with iodine excess [57].

In those areas that are iodine deficient, USI is the best strategy for the prevention and control of IDDs, since salt for household consumption is an ingredient that is used globally and in relatively constant quantities throughout the year. Additionally, the process of salt iodization is simple and efficient, which is why most countries decided to adopt USI programs (on a mandatory basis) [57,58].

However, the indicators used to monitor these programs are not always carried out permanently or systematically, which may be the reason for (at least in part) the global distribution of disorders associated with iodine intake [40,44].

Although iodine-induced hyperthyroidism and hypothyroidism can occur in people with normal thyroid function, both conditions are more frequently identified in individuals with HT, GBD, autonomous thyroid nodular disease, or in those who are under treatment with medications, such as amiodarone, lithium, and interferon-alpha, as well as in areas previously identified with long-term iodine deficiency (and which are exposed to a sudden ingestion of iodine) although such dysfunction can occur regardless of underlying thyroid autoimmunity [8,10,59].

Despite the existing association between excess iodine and the risk of hyperthyroidism or hypothyroidism, its relationship with the increase in the prevalence of TgAb and/or TPOAb positivity is less known, especially when referring to USI programs.

The studies described in Table 4 showed (in the short- to medium-term) a significant change in the increase in thyroid autoimmunity (determined by the presence of TPOAb and/or TgAb) in those areas with prolonged iodine deficiency (and which implemented USI programs); however, studies that have carried out longer-term follow-ups suggest that antibody positivity stabilizes or may even decrease. In fact, some studies have shown an increase in the prevalence of thyroid Abs before completing the first 10 years following the implementation of the USI programs. Meanwhile, other studies have found that when followed for a longer period (>15 years), the prevalence of thyroid Abs tends to decrease, reaching a frequency similar to that which existed before the USI programs were implemented [25,26,27,28,29,30,31,32,33,34,35,36,37,38,39,40,41,42,43,44,45,46,47,48,49,50,51,52,53,54,55,56,60].

Countries, such as China, have variable results in relation to the frequency of TPOAb and/or TgAb positivity depending on time. For instance, since the implementation of the USI program (more than 20 years ago), the population has been exposed to multiple iodine exposures, ranging from excessive intake (between 1996 and 2001) to more-than-adequate intake (between 2002 and 2011), and finally to adequate intake (between 2012 and 2016). Consequently, different geographic areas have intriguingly shown that the prevalence of TPOAb and/or TgAb positivity has been different following the implementation of USI programs [61].

On the other hand, in other countries, like the USA, salt fortification with iodine is voluntary (not mandatory), and the FDA does not require the iodine content to be listed on food packaging [62]. However, GBD and HT are the fourth and fifth most common autoimmune diseases in the American population, at an estimated rate (per 100,000) of 512.6 and 373.7 for GBD and HT, respectively (the median urinary iodine level in the USA is 190 µg/L) [63].

Given the distribution of HT frequency (in countries where USI programs have been established), it could be hypothesized that the prevalence of TPOAb and/or TgAb positivity (as biochemical markers of HT) varies over time depending on how USI programs are implemented, structured, maintained, and systematically evaluated.

This hypothesis can be supported by the following:The role of iodine in the induction of thyroid autoimmunity is robustly supported in animal models, e.g., in non-obese diabetic (NOD) mice, after the administration of 0.05% NaI in regular drinking water, the incidence of autoimmune thyroiditis increases to almost 100%. With the observation of chronic inflammation in the short term (3–4 weeks), along with the increase in TgAb synthesis (at the beginning) and then in TPOAb, concomitantly, the secretion of proinflammatory cytokines and cellular infiltration [mediated by T lymphocytes (TLs), B lymphocytes (BLs), and antigen-presenting cells, inter alia] are stimulated. Consequently, greater antigenicity towards Tg and TPO is produced. The depletion of regulatory TL (TReg) and the increased activity of autoreactive TL increases and amplifies the magnitude of the immune response, inducing the loss of immune tolerance and an increased risk of AITDs [2,64].In humans, there is a clear “U”-shaped relationship between iodine intake and the risk of thyroid dysfunction, whereby the body’s response to iodine deficiency is usually gradual, while in relation to excess iodine, the response can be acute or also gradual, depending on the duration and magnitude of exposure. Although acute exposure to a high iodine intake is well tolerated, it can eventually result in the presence of iodine-induced hyperthyroidism (the Jod–Basedow phenomenon) or hypothyroidism (when there is a failure to escape the Wolff–Chaikoff effect) [10,20,59].However, unlike animal models of autoimmune thyroiditis, it is actually unclear whether the immunological alterations (in patients with HT) are due to a direct effect of the iodine on the cells of the immune system (effector cells) or if, on the contrary, it is the reflection of a secondary response to the cytotoxic and/or metabolic effects on the thyroid follicular cells. In fact, sudden exposure to high amounts of iodine (in clinical scenarios where there is a known deficiency) has been associated with significant damage to thyroid tissue (mediated by the presence of free radicals) [1,2,59,65].In this regard, it has been proposed that an excess of iodine induces a greater expression of critical epitopes in Tg, increasing the probability of stimulating an autoimmune response towards the thyroid, which explains (at least in part) the increase in the prevalence of TgAb in areas where there is an excess of iodine or in those areas where USI programs have been implemented and which were exposed to a long-term iodine deficiency. In fact, it is suggested that the more severe the IDDs, the greater the incidence of thyroid autoimmunity after iodine supplementation. This concept gains more credibility if one takes into account that Tg is an autoantigen that is capable of presenting post-translational modifications as a consequence of the supply of iodine, exposing epitopes that were previously hidden [6,10,66,67].Additionally, excess iodine can have a direct impact on immune system cells that are capable of initiating and propagating the autoimmune response to the thyroid; thus, there is an increase in lymphocyte infiltration into the thyroid tissue, with a higher expression of MHC class II in thyrocytes, and an increase in the synthesis and secretion of cytokines and thyroid Abs [7,10,68].

### USI Programs, Thyroid Function, and Thyroid Autoimmunity

The results of the studies described in this review suggest that the implementation of USI programs (as an intervention strategy for IDDs) should be systematically evaluated and monitored regularly, because in those scenarios where such evaluations and monitoring are not systematically performed, the probability of IDD relapses and/or population iodine excess increases (with a higher risk of thyroid autoimmunity, which is established by an increase in the prevalence of TPOAb and/or TgAb positivity).

Despite the effects of iodine excess on thyroid function and the risk of HT, USI programs have widely demonstrated that they are the best (and most cost-effective) strategy for reducing IDDs and their global burden; however, for the implementation of these programs to be successful and to maintain the balance between a population state of “iodine sufficiency” and a low risk of IDDs or disorders associated with iodine excess and thyroid autoimmunity, they must be supported not only by close long-term supervision and monitoring, but also by the will of government entities committed to this public health problem [40,69,70] (Figure 2).

Therefore, the strategies, action plans, and various legislation of USI programs should be reviewed (and the amount of iodine in salt for human consumption should be adjusted, if necessary).

Additionally, nationally representative surveys (and sentinel studies) should be conducted periodically, addressing potential factors that may affect the normal development of USI programs, e.g., supply chain management systems, poor salt iodization practices at the population level, political will, government support, monitoring systems, continuing medical education, and public awareness of this health issue [40,57,69,70].

Furthermore, education for the population must be planned very carefully, in the sense that the consumption of iodized salt must be promoted (but in small quantities), with the aim of achieving a balance between the health benefits of consuming iodized salt and the potential risk of functional disorders, thyroid autoimmunity, high blood pressure, and the increased risk of cardiovascular outcomes, mortality, cancer, chronic kidney disease, and osteoporosis, among other undesirable outcomes [70,71,72,73].

Despite the biological plausibility between iodine excess and the presence of thyroid autoimmunity, data related to USI programs and HT are mostly incomplete, and records on baseline thyroid function and TPOAb/TgAb positivity status are not always available before USI programs are implemented (making it difficult to establish a cause–effect relationship).

It should also be noted that AITDs (and in this case, HT) are complex diseases, with multiple causes (genetic, epigenetic, environmental, and lifestyle) and triggers; therefore, iodine status would only contribute part of the total predisposing factors for its development [2,6,8,10,74]. Therefore, excess iodine (e.g., as a result of improperly implemented USI programs) can be considered a trigger for HT.

Additionally, it should be noted that a significant number of the reviewed studies focused on younger populations (schoolchildren and adolescents); therefore, and due to the fact that iodine requirements vary with age and due to different studies that have shown that AITD is more prevalent during the development of secondary sex characteristics and at puberty, these factors could have influenced the positivity rates of thyroid Abs in this population group [75,76].

## 5. Strengths and Weaknesses

This systematic review demonstrated some strengths, e.g., the comprehensive scoping review and the sensitive and robust collection of relevant data and studies from six major databases (PubMed/Medline, ProQuest, Scopus, Biosis, Web of Science, and Google Scholar), providing a broad overview of the topic.

However, several weaknesses can also be identified, e.g., the language limitation in our study selection, as well as the fact that studies evaluating other outcomes of interest (such as hypothyroidism, hyperthyroidism, goiter, and thyroid nodular disease, inter alia) were not considered. Furthermore, the methods and ways in which the diagnosis of HT was made were not the same in the studies evaluated.

Moreover, it is important to highlight that the prevalence of thyroid Ab positivity may also have changed due to the increased sensitivity and specificity of the laboratory kits used in the last two decades (in addition to other possible factors, such as overdiagnosis, changes in dietary patterns, seasonality, and diagnostic criteria and disease definition, inter alia) [15].

Therefore, the results described should only be extrapolated to populations with similar characteristics and clinical settings to those of the studies developed.

## 6. Future Implications

Long-term, multicenter, observational studies are needed in different populations and in different clinical settings, assessing not only the prevalence of thyroid Abs, but also thyroid function, goiter frequency (and glandular imaging features), and the population’s status in terms of other micronutrients (e.g., selenium, vitamin D, zinc, and iron, inter alia) and other possible factors (genetic, epigenetic, environmental, and lifestyle). Such studies are needed in order to establish the role that iodine (after the implementation of USI programs) may have on thyroid function and the risk of AITDs. Also, other sources of iodine from both food and non-food sources should be taken into account.

## 7. Conclusions

Excess iodine, mediated by poorly designed USI programs or those lacking adequate follow-up and monitoring strategies, could explain (at least in part) the prevalence and distribution of AITDs, specifically HT. Therefore, it is necessary not only to implement these programs but also to monitor them continuously to maintain low rates of IDDs and excess iodine in the population. Only a balance and collaboration between the policies and initiatives of the USI programs, together with well-designed educational campaigns and recommendations focused on reducing salt consumption, will allow for the maintenance of a low rate (or the eradication) of IDDs and a low risk of HT.

## Figures and Tables

**Figure 1 diseases-13-00166-f001:**
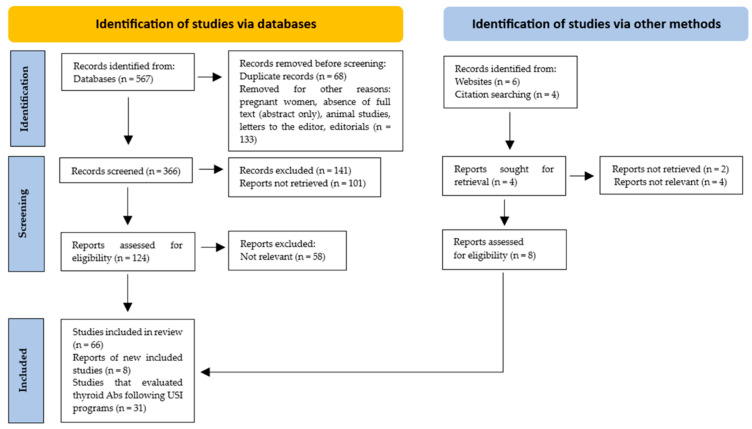
PRISMA flow diagram. Method for the selection of articles.

**Figure 2 diseases-13-00166-f002:**
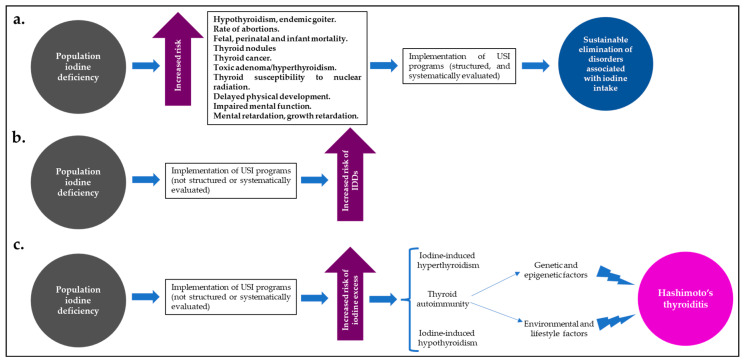
In iodine-deficient populations, well-structured and systematically evaluated USI programs (**a**) lead to a significant and sustained reduction in disorders associated with iodine intake. However, those programs that have not been well-structured or systematically evaluated may increase the risk of IDDs (**b**) or the risk of population iodine excess (**c**), with a greater likelihood of iodine-induced hyperthyroidism or hypothyroidism, or thyroid autoimmunity (HT). Abbreviations: HT: Hashimoto’s thyroiditis; USI: universal salt iodization.

**Table 1 diseases-13-00166-t001:** Inclusion criteria adopted in the systematic review.

PICO Elements	Inclusion Criteria
Population	Individuals from the general population (both, male and female)
Intervention	USI programs
Comparison	Not applicable
Outcome	Changes in the prevalence of TPOAb and/or TgAb positivity

Abbreviations: TPOAb: thyroid peroxidase antibodies; TgAb: thyroglobulin antibodies; USI: universal salt iodization.

**Table 2 diseases-13-00166-t002:** Inclusion and exclusion criteria for studies in the systematic review.

Categories	Inclusion Criteria	Exclusion Criteria
Topic	Studies correspond to iodine, salt intake and/or Hashimoto’s thyroiditis	Not applicable
Selection of databases (for searching the respective studies)	Pubmed/Medline; ProQuest; Scopus; Biosis; Web of Science, and Google Scholar	Different databases
Search limits for studies (according to time interval)	From January 1965 to January 2025	Not applicable
Population/target group	Humans	Other types of studies, for example, in animals or pregnant women
Context	Any geographic area, continent, country	Not applicable
Study design	Clinical trials, meta-analyses, reviews, scoping reviews, and systematic reviews	Other types of studies
Data extraction	Standardized template using a predefined data form (in Excel)	Other forms of data extraction
Language	English	Others

**Table 3 diseases-13-00166-t003:** Countries where prevalence studies on Hashimoto’s thyroiditis have been conducted refs. [3,4,15,16,17].

Country (and Years in Which Studies Evaluating the Prevalence of HT Were Conducted)	Study Design (and Diagnostic Criteria of HT)	% Prevalence of HT (from Lowest to Highest Reported)
Australia (2006 to 2016)	Cross-sectional [serum (Abs)]	8.6 to 13.3
Bosnia and Herzegovina (2021)	Array research [serum (Abs)]	0.43
Brazil (1995 to 2019)	Cross-sectional [serum (Abs) + TU, TU, Abs, thyroid tissue, or NR	0.1 to 19.5
China (2006 to 2021)	Cross-sectional or array research [serum (Abs), Abs + TU, or thyroid tissue]	0.3 to 16.1
Colombia (2023)	Cross-sectional [serum (Abs)]	22.3
Croatia (2022)	Cross-sectional [serum (Abs)]	23.7
Denmark (2003 to 2024)	Cross-sectional [serum (Abs)]	18 to 39.7
England (1966 to 1990)	Cross-sectional [serum (Abs)]	2.0 to 17.8
Finland (1971 to 1972)	Cross-sectional [serum (Abs)]	7.8 to 9.5
Germany (2003 to 2016)	Cross-sectional [serum (Abs)]	1.2 to 14
Ghana (2017)	Retrospective cohort [serum (Abs + TU)]	7.2
Iran (2017)	Cross-sectional [serum (Abs)]	12.8
Italy (1999 to 2019)	Cross-sectional [serum (Abs)]	2.6 to 35.1
Japan (1991 to 2007)	Cross-sectional [serum (Abs), thyroid tissue, or Abs + TU + FNA]	1.0 to 18
Jordan (2022)	Cross-sectional [serum (Abs)]	15.1
Mexico (2015)	Cross-sectional [serum (Abs) + TU]	8.4
Nigeria (2007)	Cross-sectional [serum (Abs)]	6.7
Norway (1984 to 1996)	Cross-sectional [serum (Abs)]	3.4 to 6.9
Poland (2017)	Cross-sectional [serum (Abs)]	5.0
Russia (2021)	Cross-sectional [serum (Abs)]	0.42
South Korea	Array research [NR]	0.1
Spain (2017)	Cross-sectional [serum (Abs)]	8.6
Sri Lanka (2012)	Cross-sectional [serum (Abs)]	6.8
Tunisia (2006)	Array research [serum (Abs)]	22.8
USA (1994 to 2024)	Cross-sectional [serum (Abs), thyroid tissue, or NR]	0.4 to 22.4

Abbreviations: Abs: antibodies; FNA: fine needle aspiration; NR: not reported; TU: thyroid ultrasound.

**Table 4 diseases-13-00166-t004:** Population iodine status in countries where Hashimoto’s thyroiditis prevalence studies have been conducted, adapted from refs. [8,13,21,24].

Country	Median (mUIC, µg/L)	Date of Survey (Source)	Population Surveyed (Age)	Iodine Intake
Australia	175	2011–2012 (N)	SAC (5–11)	Adequate
Bosnia and Herzegovina	157	2005 (N)	SAC (7–10)	Adequate
Brazil	276	2016 (N)	SAC (6–14)	Adequate
China	200	2017 (N)	SAC (9–11)	Adequate
Colombia	407	2015–2016 (N)	SAC (5–12)	Excessive
Croatia	248	2009 (N)	SAC (7–10)	Adequate
Denmark	145	2015 (S)	SAC	Adequate
England (United Kingdom)	149	2016/17–2018/19 (N)	SAC (4–10)	Adequate
Finland	96	2017 (N)	Adults (25–74)	Insufficient
Germany	89	2014–2017 (N)	SAC (3–17)	Insufficient
Ghana	130	2011 (N)	SAC (6–12)	Adequate
Iran	186	2016–2017	SAC (8–10)	Adequate
Italy	118	2015–2019 (S)	SAC	Adequate
Japan	265	2013–2017 (N)	SAC (6–12)	Adequate
Jordan	203	2010 (N)	SAC (8–10)	Adequate
Mexico	297	2011 (N)	SAC (6–12)	Adequate
Nigeria	130	2004–2005 (N)	SAC (9–12)	Adequate
Norway	75	2017–2018 (S)	WRA (18–30)	Insufficient
Poland	112	2009–2011 (S)	SAC (6–12)	Adequate
Russia	<100	2008–2020 (S)	SAC	Insufficient
South Korea	449	2013–2015 (N)	SAC (6–19)	Excessive
Spain	173	2011–2012 (N)	SAC	Adequate
Sri Lanka	233	2016 (N)	SAC (6–12)	Adequate
Tunisia	220	2013 (N)	SAC (6–12)	Adequate
USA	190	2011–2014 (N)	SAC (6–11)	Adequate

Abbreviations: mUIC: median urinary iodine level; N: nationally representative survey; S: sub-national survey; SAC: school-age children; WRA: women of reproductive age.

**Table 5 diseases-13-00166-t005:** Studies that have evaluated changes in TPOAb and/or TgAb levels following the mandatory implementation of USI programs. Refs. [25,26,27,28,29,30,31,32,33,34,35,36,37,38,39,40,41,42,43,44,45,46].

Author; Year (Ref.)	Study Population (n = Number of Participants)	Country and Evaluation Post-USI (Years)	Changes in the Prevalence of TPOAb and/or TgAb
Premawardhana LDKE, et al., 2000 [47].	Female schoolchildren [aged 11–16 years] from areas with different endemic goiter prevalence (367)	Sri Lanka (5)	TgAb prevalence of 42.1%; TPOAb prevalence of 8.7% in all ages (after USI); 14.3% at 11 years; 19.5% at 12 years; 44.1% at 13 years; 53% at 14 years; 52% at 15 years, and 69.7% in 16-year-old schoolchildren
Azizi F, et al., 2002 [48].	1323 people aged 3 to 70 years (in 1983) and 3146 people aged 3 to 70 years (in 1995)	Iran (12)	Positive thyroid Abs were present in 3.1 and 3.2% of cases in 1983, and in 1.9 and 1.9% of cases in 1995 for TPOAb and TgAb, respectively. The prevalence of positive thyroid Abs in females >18 years of age was 9.4% and 5.2% in 1983 and 1995, respectively.
Mazzioti G, et al., 2003 [49].	Female schoolchildren [aged 11–17.5 years] from three areas with different endemic goiter prevalence [follow-up of 42 schoolchildren, 3 years later] (282)	Sri Lanka (8)	TgAb prevalence of 34.8%; TgAb + TPOAb prevalence of 46.9%; reduced TgAb prevalence (from ≥70% to about 40%). Increased TPOAb prevalence (from <10% to 18.6%). Reduced thyroid Abs prevalence (TgAb + TPOAb 23.8% vs. 46.9%)
Zimmermann MB, et al., 2003 [50].	b. Iodine-deficient schoolchildren [aged 6–15 years]	Morocco [1]	The prevalence of elevated Abs titers was low (only 1% had elevated TPOAb before and after introduction of iodine, and no child had an elevated TgAb during the study period)
Zois C, et al., 2003 [51].Fountoulakis S, et al., 2007 [52].	c. Schoolchildren [aged 12–18 years] in a formerly iodine-deficient community (302)	Greece (7)	TPOAb and TgAb prevalence was 8.3% and 5.6%, respectively. Both thyroid Abs antibodies were positive in 3.3% of cases. The prevalence of autoimmune thyroiditis increased from 3.3% to 9.6%
Pedersen IB, et al., 2003 [53].	d. Adult general population [aged 18–65 years], from two areas of with mild and moderate iodine deficiency (4184)	Denmark (0–1)	TPOAb and/or TgAb prevalence was 18.8%; TPOAb or TgAb prevalence was 13.1% or 13.0%, respectively. TPOAb and TgAb prevalence was 7.3%
Laurberg P, et al., 2006 [54].	Community-dwelling population sampled from two areas with different iodine intakes. Adult population (4649)	Denmark (4–5)	The overall prevalence of one or both antibodies was 18.8%. TPOAb prevalence 13.1%; TgAb 13.0%
Teng W, et al., 2006 [55].	Three representative communities with different levels of iodine intake. Aged >13 years (3018)	China (5)	The overall prevalence of TPOAb and TgAb positivity in the three regions was 9.8% and 9.1%, respectively (with no significant differences between these regions)
Bastemir M, et al., 2006 [56].	Two regions with different iodine status after two years of iodization. A total of 1733 adolescent subjects were enrolled into the study (993 from an iodine-sufficient area (group 1) and 740 from an iodine-deficient area (group 2).	Turkey (2)	The percentage of TgAb-positive subjects was found to be 17.6% in group 1 and 6.4% in group 2; that of TPOAb-positive subjects was 4.3% in group 1 and 1.5% in group 2. The prevalence of TgAb and/or TPOAb positivity was higher in group 1 than in group 2 (18.52% vs. 6.62%, respectively)
Heydarian P, et al., 2007 [25].	Random cluster sampling from the adult population (1426)	Iran (5–6)	Positive TPOAb and/or TgAb were detected in 22.2% of cases, while positivity of both TPOAb and TgAb was present in 7% of cases. Compared to non-goitrous females, goitrous females had a frequency of positive TPOAb of 22.7 vs. 12.5%, respectively
Gołkowski F, et al., 2007 [26].	1424 adults (≥16 years) with negative medical history for thyroid disorders, from an area with moderate iodine deficiency	Poland (8–10)	Increase in the serum concentration of TPOAb (4.9% to 12.1%)
Li Y, et al., 2008 [27].	Individuals aged ≥13 years from three communities with differentlevels of iodine intake; baseline study: females (2827); males (934). Follow-up study: females (1748); males (633)	China (3–8)	TPOAb and TgAb (more frequent in women and in areas with higher I intake) was 9.81 and 9.09%, respectively. Follow-up cumulative incidence was TPOAb 2.92%; TgAb 3.87%
Aminorroaya A, et al., 2010 [28].	Adult population (2523)	Iran (15)	TPOAb and TgAb were positive in 29.2% and 29.4% of cases, respectively. Positive TPOAb was present in 24% of the non-goitrous and 33.5% of goitrous subjects. TgAb was positive in 21.6% of the non-goitrous and 35.9% of the goitrous subjects
Pedersen IB, et al., 2011 [29].	Community-dwelling population sampled from two areas with different iodine intake; females [19–65 years], (3712); males [61–65 years], (937)	Denmark (4–5)	TPOAb prevalence was 14.3% before USI and increased to 23.8% after; TgAb prevalence was 13.9% and increased to 19.9%
Marwaha RK, et al., 2012 [30].	Schoolchildren [5–18 years of age) from 25 schools located in 16 regions from 5 geographical zones (38,961)	India (20)	TPOAb was positive in 3.6% and strongly positive in 1.8% of children. It was seen to increase with increasing age in children. Girls had a higher prevalence of TPOAb positivity than boys (5.1% vs. 2.3%)
Marwaha RK, et al., 2012 [31].	Adult members of resident welfare associations of 5 residential colonies (4409). Adult population, from 18–90 years of age	India (20)	TPOAb was positive in 13.3% of adults and showed a positive correlation with age, female sex, and hypothyroidism
Fernando RF, et al., 2012 [32].	National study on epidemiology and prevalence of goiters. A total of 5200 individuals were screened, and 426 were clinically detected as having goiters. The sample selected for antibody testing totaled 153	Sri Lanka (12)	TPOAb prevalence of 41.8% (among patients with goiter)
Aghini Lombardi F, et al., 2013 [33].	General community survey; 1148 residents were examined: 83 (39 males and 44 females) 1–14-year-old subjects, and 1065 (429 males and 636 females) aged ≥ 15 years.	Italy (15)	The frequency of positive thyroid Abs was significantly higher in 2010 (19.5%) than in 1995 (12.6%) both in females (25.6% vs. 17.2%, respectively) and in males (10.7% vs. 5.8%, respectively)
Miranda DM, et al., 2015 [34].	Schoolchildren [7–14 years old) in two distinct periods of time in which fortified salt had different concentrations of iodine (206)	Brazil (10)	TPOAb prevalence of 1.0% and 5.5% in boys and girls, respectively; TgAb prevalence of 1.0% and 3.6% in boys and girls, respectively
Shan Z, et al., 2016 [35].	Participants from 10 cities, ≥15 years old (15,008). The 10 city cohorts were similar in age and sex but differed in iodine intake.	China (16)	TPOAb and TgAb prevalence in the whole cohort population was 11.5% and 12.0%, respectively, with a higher prevalence in women than in men (14.8% vs. 7.0% and 18.1% vs. 5.1% for TPOAb and TgAb, respectively). TPOAb and TgAb prevalence also increased significantly (11.5% vs. 9.81%, for TPOAb and 12.6% vs. 9.09%, for TgAb, respectively), after the implementation of the USI programs
Khattak RM, et al., 2016 [36].	Population-based data from the same study region (4308 and 4420 subjects, respectively), and according to the follow-up period, 1997–2001 and 2008–2012, respectively. Adult population	Germany (7 and 17)	TPOAb positivity decreased from 3.9% to 2.9% (in the total population), from 1.3% to 1.4%, and from 6.7% to 4.7% (in women and men, respectively)
Hong A, et al., 2017 [37].	The major primary care and largest public hospital pathology providers (389,910). Adult population	Tasmania (18)	There was no significant change in the overall percentage of TPOAb-positive results (18.6% vs. 21.6%, in participants <40 years old), and TPOAb prevalence was 28.7% vs. 28.1% in participants >40 years old, before and after the implementation of the USI
Bonofiglio D, et al., 2017 [38].	Participants from an iodine-deficient rural (274) and an iodine-sufficient urban area (286)	Italy (10)	The prevalence of TgAb in urban and rural areas was close to 30% and 18%, respectively. On the other hand, the prevalence of TPOAb in both urban and rural areas was close to 2.5%.
Chen X, et al., 2019 [39].	Participants of 10 cities. Adult population (14,230)	China (>10 years)	The prevalence of positivity of TgAb was 5.5% and that of TPOAb was 4.4%, while the positivity of both (TgAb and TPOAb) was 7.2%
Vargas–Uricoechea H, et al., 2019 [40].	Schoolchildren from urban areas (140)	Colombia (USI program from 1947)	Positive TPOAb was found in 42.75% of the participants; 2.87% presented positivity for TgAb and 3.62% were positive for both
Wan S, et al., 2020 [41].	Participants from three regions with different water iodine contents. Adult population (1225)	China (25)	TPOAb prevalence of 11%, 7,7%, and 10.9% (in iodine-deficient areas, iodine-adequate areas, and iodine-excess areas, respectively). TgAb prevalence of 19.1%, 10.5%, and 11.3% (in iodine-deficient areas, iodine-adequate areas, and iodine-excess areas, respectively).
Teng D, et al., 2020 [42].	78,470 subjects (≥18 years old) from 31 provincial regions of mainland China.	China (20)	Positive TPOAb and TgAb were detected in 10.19% and 9.70% of the participants, respectively. The prevalence of positive isolated TPOAb, positive isolated TgAb, and double positive TPOAb and TgAb was 4.52%, 4.16%, and 5.94%, respectively. The prevalence of isolated TPOAb was inversely associated with more than adequate iodine intake and excessive iodine intake
Jayatissa R, et al., 2021 [43].	Children and adolescents [10 to 18 years old] of both sexes (882)	Sri Lanka (>20)	TPOAb prevalence of 10.3%; TgAb prevalence of 6.4%
Vargas–Uricoechea H, et al., 2022 [44].	Participants of urban and rural areas, from four geographic regions. Healthy adult population (412)	Colombia (USI program from 1947)	TPOAb prevalence of 18.2%; TgAb prevalence of 10%. The prevalence of TPOAb and TgAb positivity was higher in participants ≥60 years
Li J, et al., 2023 [45].	Nationwide epidemiological survey. Adult population (78,470)	China (20)	The prevalence of thyroid antibody positivity (according to the presence of euthyroidism, mild subclinical hypothyroidism, severe subclinical hypothyroidism, and overt hypothyroidism) was 6.7%, 18.3%, 41%, and 51.8% (for TPOAb) and 6.8%, 17.4%, 34%, and 47.4% (for TgAb), respectively.
Fan X, et al., 2023 [46].	Community residents, permanent residents who have lived in the region for more than 5 years. Adult population (2650)	China (10)	TPOAb positivity was 6.26% and 13.5%, and TgAb positivity was 4.8% and 13.9% (in men and women, respectively). TPOAb positivity was 10.5% and 8.9%, and TgAb positivity was 9.9% and 8.4% (in urban and rural areas, respectively)

Abbreviations: TPOAb: thyroid peroxidase antibodies; TgAb: thyroglobulin antibodies; USI: universal salt iodization.

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
