# Peer review of "Iodine Intake from Universal Salt Iodization Programs and Hashimoto’s Thyroiditis: A Systematic Review"

_diseases, 2025, doi:10.3390/diseases13060166_

Round 1
Reviewer 1 Report
Comments and Suggestions for Authors
Comments to the Authors:
- A well structured review with a thorough understanding of current iodine deficiency disorders issue. A lack of a committed review process after USI implementation is common in this field and advocacy is a must in order to bring success and benefits of USI and iodine nutrition to the community.
- Figure 2, Line 5, Page 13/18: Confusion exists between ‘thyroid autoimmunity (HT)’ and HT: Hashimoto’s thyroiditis is just a typo error?
- Recently China has revised their salt concentration in USI from 40 to 60 mg/Kg (1995) to 20 to 50 mg/Kg (2012 to current) because there were a few areas identified having very high iodine in their drinking water. Would this change of standard affecting your interpretation?
Author Response
Please see the attachment
Author's Reply to the Review Report (Reviewer 1)
Manuscript ID: diseases-3638586.
Type of manuscript: Systematic Review.
Title: Iodine Intake From Universal Salt Iodization Programs and Hashimoto's thyroiditis: A Systematic Review.
Reviewer 1:
Comments and Suggestions for Authors
Comments to the Authors:
- Reviewer 1: A well structured review with a thorough understanding of current iodine deficiency disorders issue. A lack of a committed review process after USI implementation is common in this field and advocacy is a must in order to bring success and benefits of USI and iodine nutrition to the community.
Answer: Many thanks to the reviewer for their comment; we are honored by their opinion of our manuscript.
- Reviewer 1: Figure 2, Line 5, Page 13/18: Confusion exists between ‘thyroid autoimmunity (HT)’ and HT: Hashimoto’s thyroiditis is just a typo error?
Answer: Many thanks to the reviewer for the comment, the legend of figure 2 says the following:
Figure 2. In iodine–deficient populations, well–structured and systematically evaluated USI programs (a) lead to a significant and sustained reduction in disorders associated with iodine intake. However, those programs that have not been well–structured or systematically evaluated may increase the risk of IDDs (b) or the risk of population iodine excess (c), with a greater likelihood of iodine–induced hyperthyroidism or hypothyroidism, or thyroid autoimmunity (HT). Abbreviations: HT: Hashimoto’s thyroiditis; USI: universal salt iodization.
In this regard, we want to emphasize (when referring to thyroid autoimmunity) specifically Hashimoto's thyroiditis (HT). Later, in the same figure legend, we have simply included the abbreviation for Hashimoto's thyroiditis (HT). Therefore, we confirm that it is not a typographical error.
- Reviewer 1: Recently China has revised their salt concentration in USI from 40 to 60 mg/Kg (1995) to 20 to 50 mg/Kg (2012 to current) because there were a few areas identified having very high iodine in their drinking water. Would this change of standard affecting your interpretation?
Answer: Many thanks to the reviewer for the question; we completely agree; this is why we refer to it in the manuscript (in the discussion section). The paragraph reads as follows:
“Countries such as China have variable results in relation to the frequency of TPOAb and/or TgAb positivity depending on time. For instance, since the implementation of the USI program (more than 20 years ago), the population has been exposed to multiple iodine exposures, ranging from excessive intake (between 1996 and 2001) to more-than-adequate intake (between 2002 and 2011), and finally to adequate intake (between 2012 and 2016). Consequently, different geographic areas have intriguingly shown that the prevalence of TPOAb and/or TgAb positivity has been differential following the implementation of USI programs [61]”.
In this sense, China has historically gone through several classification definitions of its population iodine status (insufficiency, excessive, and adequate); Therefore, this ongoing change in the policies and strategies of USI programs in China can only be determined by monitoring and controlling them over time. In fact, Table 5 describes the five population-based studies conducted in China, which demonstrate the different changes in iodine status since the implementation of USI. Finally, several countries with USI programs have been modifying the standard of salt iodization for human consumption based on the median urinary iodine of the population studied (or based on the iodine content in drinking water or other consumer products; e.g., flour, milk, inter alia).
We hope we've been able to adequately respond to and clarify the reviewer's suggestions, questions, and comments; thank you again for the feedback. We'll be attentive to any additional corrections that may be required.
Cordially:
Hernando Vargas-Uricoechea.
First author and corresponding author.
Reviewer 2 Report
Comments and Suggestions for Authors
Comments to the Authors
Vegas-Uricoechea and colleagues have conducted a thorough review of the incidence of Hashimoto’s thyroiditis, using the positivity rates of anti-thyroid antibodies (TgAb/TPOAb) before and after the implementation of salt iodization programs. While this approach provides valuable insights, it is important to acknowledge that the primary objective of global salt iodization initiatives was to address iodine deficiency and its associated thyroid dysfunctions, particularly hypothyroidism. Therefore, incorporating data on thyroid function, specifically hypothyroidism, would offer a more comprehensive understanding of the program's impact.
The authors concluded that inadequately monitored salt iodization programs may lead to excessive iodine intake, potentially contributing to the development of Hashimoto’s thyroiditis. While this hypothesis warrants consideration, it is essential to recognize that salt iodization has been instrumental in mitigating iodine deficiency-related disorders worldwide. Notably, countries like South Korea and Japan did not implement mandatory iodization programs due to their established iodine-rich food cultures. Additionally, due to the vast land, China has iodine nutrition issues that vary significantly across regions, from mountainous to coastal areas. Direct comparisons of those countries with nations requiring iodized salt must be complicated.
Specific Comments
- As previously mentioned, the primary aim of salt iodization programs was to combat iodine deficiency and associated thyroid dysfunctions, particularly hypothyroidism. Therefore, it is recommended that the review incorporate data on thyroid function, especially hypothyroidism, to provide a more comprehensive analysis.
- On page 2, lines 61-62, the manuscript refers to iodine-induced hyperthyroidism as an autoimmune condition. However, this phenomenon, known as the Jod-Basedow, typically occurs in individuals with pre-existing thyroid toxic nodular goiter, and is not classified as an autoimmune disorder. It would be beneficial to rephrase this section to accurately reflect the pathophysiology.
- On page 5, line 160, the manuscript discusses the prevalence of Hashimoto’s thyroiditis. It is important to note that prevalence can be influenced by various factors, including the geographical area and the precision of the testing kits used. Given the study period spans from 1966 to 2024, advancements in diagnostic technology may have improved the sensitivity and specificity of antibody detection, potentially affecting reported prevalence rates.
- Table 5 indicates that many of the reviewed studies focused on younger populations. Considering that iodine requirements vary with age, and that puberty involves significant hormonal changes, these factors could influence antibody positivity rates. Please discuss this point.
- On page 11, lines 230-233, the manuscript addresses drug-induced thyroid dysfunction. It is important to clarify that such dysfunction can occur regardless of underlying thyroid autoimmunity. Rephrasing this section would help prevent potential misunderstandings regarding the relationship between drug-induced thyroid dysfunction and pre-existing autoimmune conditions.
Author Response
Author's Reply to the Review Report (Reviewer 2)
Manuscript ID: diseases-3638586.
Type of manuscript: Systematic Review.
Title: Iodine Intake From Universal Salt Iodization Programs and Hashimoto's thyroiditis: A Systematic Review.
Reviewer 2:
Comments and Suggestions for Authors
Comments to the Authors
Vargas-Uricoechea and colleagues have conducted a thorough review of the incidence of Hashimoto’s thyroiditis, using the positivity rates of anti-thyroid antibodies (TgAb/TPOAb) before and after the implementation of salt iodization programs. While this approach provides valuable insights, it is important to acknowledge that the primary objective of global salt iodization initiatives was to address iodine deficiency and its associated thyroid dysfunctions, particularly hypothyroidism. Therefore, incorporating data on thyroid function, specifically hypothyroidism, would offer a more comprehensive understanding of the program's impact.
The authors concluded that inadequately monitored salt iodization programs may lead to excessive iodine intake, potentially contributing to the development of Hashimoto’s thyroiditis. While this hypothesis warrants consideration, it is essential to recognize that salt iodization has been instrumental in mitigating iodine deficiency-related disorders worldwide. Notably, countries like South Korea and Japan did not implement mandatory iodization programs due to their established iodine-rich food cultures. Additionally, due to the vast land, China has iodine nutrition issues that vary significantly across regions, from mountainous to coastal areas. Direct comparisons of those countries with nations requiring iodized salt must be complicated.
Specific Comments
- Reviewer 2: As previously mentioned, the primary aim of salt iodization programs was to combat iodine deficiency and associated thyroid dysfunctions, particularly hypothyroidism. Therefore, it is recommended that the review incorporate data on thyroid function, especially hypothyroidism, to provide a more comprehensive analysis.
Answer: Many thanks to the reviewer for the comment and suggestion. We fully agree. However, at the time of the systematic search for studies (and their selection), we found that the vast majority of these studies did not consider thyroid functional outcomes (hypothyroidism or hyperthyroidism). For example, many of them evaluated specific outcomes (such as goiter rate, thyroid ultrasound findings, thyroglobulin levels, and urinary iodine levels); others only evaluated TSH values ​​(without FT4 or FT3), which did not allow for adequate classification or stratification of the diagnosis of underlying thyroid dysfunction). Additionally, the few studies that did evaluate thyroid functional outcomes (the majority) did not describe the criteria for defining the disease (hypothyroidism or hyperthyroidism) before or after the implementation of the USI programs. These aspects made it difficult to “add” a column or table in which thyroid functional outcomes could be summarized; therefore, we focused (in Table 5) on describing the outcomes on thyroid autoimmunity, based on the prevalence of TPOAb and TgAb positivity.
- Reviewer 2: On page 2, lines 61-62, the manuscript refers to iodine-induced hyperthyroidism as an autoimmune condition. However, this phenomenon, known as the Jod-Basedow, typically occurs in individuals with pre-existing thyroid toxic nodular goiter, and is not classified as an autoimmune disorder. It would be beneficial to rephrase this section to accurately reflect the pathophysiology.
Answer: Many thanks to the reviewer for the concept and suggestion. We fully agree. Therefore, we have rewritten the aforementioned paragraph and modified the order of the references (number 14) to support the concept outlined (and we have highlighted the changes in yellow).
The paragraph reads as follows:
“In the presence of excessive iodine intake or exposure, hypothyroidism or hyperthyroidism may occur (when the intrinsic regulatory mechanisms of protection against excess iodine fail). Furthermore, sustained exposure to excess iodine has also been linked to multiple other outcomes; however, studies have not been able to reproduce these outcomes in the following ways: diabetes, hypertension, cardiovascular mortality, papillary thyroid carcinoma, among others”
- Reviewer 2: On page 5, line 160, the manuscript discusses the prevalence of Hashimoto’s thyroiditis. It is important to note that prevalence can be influenced by various factors, including the geographical area and the precision of the testing kits used. Given the study period spans from 1966 to 2024, advancements in diagnostic technology may have improved the sensitivity and specificity of antibody detection, potentially affecting reported prevalence rates.
Answer: Many thanks to the reviewer for the concept and suggestion. We fully agree. Therefore, we have inserted a paragraph on page 14, in the “Strengths and Weaknesses” section, which reads as follows:
“Moreover, it is important to highlight that the prevalence of thyroid Ab positivity may have also changed due to the increased sensitivity and specificity of laboratory kits used in the last two decades (in addition to other possible factors such as overdiagnosis, changes in dietary patterns, seasonality, diagnostic criteria and disease definition, inter alia)”.
We also cite reference number 15 to support the concept outlined in that paragraph, and we have highlighted the changes in yellow.
- Reviewer 2: Table 5 indicates that many of the reviewed studies focused on younger populations. Considering that iodine requirements vary with age, and that puberty involves significant hormonal changes, these factors could influence antibody positivity rates. Please discuss this point.
Answer: Many thanks to the reviewer for the concept and suggestion; we completely agree. Several studies describe exactly the same thing the reviewer noted. Therefore, we have inserted a paragraph at the end of page 13 (and immediately before "Strengths and Weaknesses") that reads:
“Additionally, it should be noted that a significant number of the reviewed studies focused on younger populations (schoolchildren and adolescents); therefore, and due to the fact that iodine requirements vary with age and due to different studies that have shown that AITD is more prevalent during the development of secondary sex characteristics and at puberty, these factors could have influenced the positivity rates of thyroid Abs in this population group”.
We have also inserted a couple of references [75, 76] to support the concept expressed in that paragraph and we have highlighted the changes in yellow.
- Reviewer 2: On page 11, lines 230-233, the manuscript addresses drug-induced thyroid dysfunction. It is important to clarify that such dysfunction can occur regardless of underlying thyroid autoimmunity. Rephrasing this section would help prevent potential misunderstandings regarding the relationship between drug-induced thyroid dysfunction and pre-existing autoimmune conditions.
Answer: Many thanks to the reviewer for the suggestion, we completely agree, therefore, we have inserted the concept as recommended by the reviewer, which was as follows: “although such dysfunction can occur regardless of underlying thyroid autoimmunity”.
- We have highlighted this change in yellow.
Once again, many thanks to the reviewer for their suggestions and comments, which undoubtedly allow us to improve our manuscript.
We hope we have fully addressed these suggestions; we will be attentive to any corrections or changes that may be necessary.
Cordially:
Hernando Vargas-Uricoechea.
First author and corresponding author.
Reviewer 3 Report
Comments and Suggestions for Authors
This is a well written and clinically sound paper.
The reference section seems updated, although no trace is found of previous similar reviews (on pubmed, some in the last two/three years). These should be considered, added and discussed.
Minor concerns
on page 3 (PRISMA diagram) Identification (second column), the authors state: "Removed for other reasons = 133", specify the reasons
Table 2 caption is unnecessary (no abbreviations in the table text)
Table 2: in exclusion criteria "aplicable" is wrong and should be amended "applicable"
Additional meta-analysis on the papers retrieved should be done, since this would add interest to the paper
Author Response
Author's Reply to the Review Report (Reviewer 3)
Manuscript ID: diseases-3638586.
Type of manuscript: Systematic Review.
Title: Iodine Intake From Universal Salt Iodization Programs and Hashimoto's thyroiditis: A Systematic Review.
Reviewer 3:
Comments and Suggestions for Authors
- Reviewer 3: This is a well written and clinically sound paper.
Answer: Many thanks to the reviewer for their comment; we are honored by their opinion of our manuscript.
- Reviewer 3: The reference section seems updated, although no trace is found of previous similar reviews (on pubmed, some in the last two/three years). These should be considered, added and discussed.
Answer: Many thanks to the reviewer for the suggestion. In reality, there are very few studies that evaluate and analyze iodine intake (based on USI programs) and the risk of developing Hashimoto's thyroiditis. This was the fundamental reason for conducting this systematic review, as we considered it necessary to conduct a comprehensive review and synthesize the information.
In this systematic and analytical literature search, we reviewed and included in the references those that had previously addressed the topic (e.g., references 8, 13, 14, 15, and 60, all from 2024 and 2025).
Thus (and given the limited literature that condensed the information we wanted to convey), we turned to the original studies and summarized them in the manuscript using the respective tables. Furthermore, in the discussion, we took into account some of the concepts outlined in these reviews, but we tried to draw conclusions based on the findings described in the original articles.
Minor concerns
- Reviewer 3: On page 3 (PRISMA diagram) Identification (second column), the authors state: "Removed for other reasons = 133", specify the reasons.
Answer: Many thanks to the reviewer for pointing out this detail in the figure. We fully agree; therefore, we have inserted what was suggested. We will also submit the new version of the figure (although we have already inserted it in the manuscript).
- Reviewer 3: Table 2 caption is unnecessary (no abbreviations in the table text).
Answer: Many thanks to the reviewer for the suggestion; we fully agree; therefore, we have replaced the initial title with the following: Inclusion and exclusion criteria for studies in the systematic review.
The above is to provide context for the interested reader. Finally, we highlight the change (in the manuscript) in yellow.
Reviewer 3: Table 2: in exclusion criteria "aplicable" is wrong and should be amended "applicable".
Answer: Many thanks to the reviewer for identifying this detail in the table; we fully agree, so we've made the changes and highlighted them in yellow. This detail also appeared in Table 1 and has also been corrected and highlighted in yellow.
Reviewer 3: Additional meta-analysis on the papers retrieved should be done, since this would add interest to the paper.
Answer: Many thanks to the reviewer for the suggestion; in fact, we completely agree. However, on lines 130 to 133 (in section 2.3. Data analysis), we describe why we didn't conduct a meta-analysis. The paragraph reads as follows:
"No statistical analysis or meta–analysis was performed due to the high heterogeneity observed among the studies included in this review. However, we developed a descriptive analysis to summarize and synthesize the most important characteristics of the selected studies (choosing to adopt a narrative approach)”.
Once again, many thanks to the reviewer for their suggestions and comments, which undoubtedly allow us to improve our manuscript.
We hope we have fully addressed these suggestions; we will be attentive to any corrections or changes that may be necessary.
Cordially:
Hernando Vargas-Uricoechea.
First author and corresponding author.
Round 2
Reviewer 2 Report
Comments and Suggestions for Authors
Comments to the authors
The authors have made significant improvements to the manuscript. However, some data are still missing. Including information on thyroid function—at least TSH levels—would further enhance the quality of the manuscript.
As these data are currently unavailable, I have no further comments for the authors at this time.
Reviewer 3 Report
Comments and Suggestions for Authors
I believe the authors have substantially answered to the possible concerns raised during the first round of review.
The paper should be accepted for publication after the usual copy editing process, in my opinion.